# Formulation and Evaluation of Prednisolone Sodium Metazoate-Loaded Mucoadhesive Quatsomal Gel for Local Treatment of Recurrent Aphthous Ulcers: Optimization, In Vitro, Ex Vivo, and In Vivo Studies

**DOI:** 10.3390/pharmaceutics15071947

**Published:** 2023-07-14

**Authors:** Ashraf Kassem, Hanan Refai, Mohamed A. El-Nabarawi, Menna M. Abdellatif

**Affiliations:** 1Department of Pharmaceutics, College of Pharmaceutical Sciences and Drug Manufacturing, Misr University for Science and Technology, Giza 12566, Egypt; ashraf.kassam@must.edu.eg; 2Department of Pharmaceutics and Industrial Pharmacy, Faculty of Pharmacy, Cairo University, El-Kasr El-Aini Street, Cairo 11562, Egypt; mohamed.elnabarawi@pharma.cu.edu.eg; 3Department of Industrial Pharmacy, College of Pharmaceutical Sciences and Drug Manufacturing, Misr University for Science and Technology, Giza 12566, Egypt; menna.abdallatif@must.edu.eg

**Keywords:** recurrent aphthous ulcer, buccal mucoadhesive gel, quatsomes, prednisolone sodium metazoate

## Abstract

This study aims to formulate a buccal mucoadhesive gel containing prednisolone sodium metazoate-loaded quatsomes for efficient localized therapy of recurrent aphthous ulcers. Quatsomes were prepared using a varied concentration of quaternary ammonium surfactants (QAS) and cholesterol (CHO). A 2^3^ factorial design was conducted to address the impact of independent variables QAS type (X_1_), QAS to CHO molar ratio (X_2_), and sonication time (X_3_). The dependent variables were particle size (PS; Y_1_), polydispersity index (PDI; Y_2_), zeta potential (ZP; Y_3_), entrapment efficiency percent (EE%; Y_4_) and percent of drug released after 6 h (Q6%: Y_5_). Then, the selected quatsomes formula was incorporated into different gel bases to prepare an optimized mucoadhesive gel to be evaluated via in vivo study. The PS of the developed quatsomes ranged from 69.47 ± 0.41 to 113.28 ± 0.79 nm, the PDI from 0.207 ± 0.004 to 0.328 ± 0.004, ZP from 45.15 ± 0.19 to 68.1 ± 0.54 mV, EE% from 79.62 ± 1.44 to 98.60% ± 1.22 and Q6% from 58.39 ± 1.75 to 94.42% ± 2.15. The quatsomal mucoadhesive gel showed rapid recovery of ulcers, which was confirmed by the histological study and the evaluation of inflammatory biomarkers. These results assured the capability of the developed quatsomal mucoadhesive gel to be a promising formulation for treating buccal diseases.

## 1. Introduction

Recurrent aphthous ulcers (RAUs), also known as recurrent aphthous stomatitis, is a prevalent condition that induces ulcers in the oral mucosa; it also decreases the quality of life as patients experience difficulty in masticating, talking, and swallowing [1]. RAU is an oral mucosal lesion ranging from less than 1 mm to more than 1 cm. It is distinguished by various, more often-occurring oval-shaped ulcers with a yellowish pseudomembrane on the ulcer floor and an erythematous border [2]. With a 50% recurrence rate after three months, these ulcers can affect up to 25% of the general population [3]. The etiopathogenesis of this condition is still unknown; however, it is thought to be multifactorial. Various etiologies of RAU include food allergies, hormonal and gastrointestinal disorders, viral and bacterial infections, mechanical injuries, and stress [4,5,6]. The diagnosis, clinical features, and severity of RAU affect treatment decisions. The goal of therapy includes pain control, suppression of the inflammatory response, reduction of recurrence frequency, reduction of ulcer border, and avoiding harmful effects of systemic treatments. RAU treatment includes topical and systemic drugs such as corticosteroids, antiseptics, and antibacterials, which are used singly or in varied combinations [7,8,9].

Corticosteroids are often recommended as first-line treatment for RAU [10]; however, the efficacy of topical corticosteroids is limited for treating deep and large ulcers; in such cases, systemic corticosteroids are commonly used. However, numerous patients cannot tolerate the side effects of systematic corticosteroids [11].

Prednisolone sodium metazoate (Pred MS) is a synthetic glucocorticoid with anti-inflammatory and immunomodulating properties. It also decreases the number of circulating lymphocytes and induces cell differentiation. Moreover, Pred MS provides topical or local anti-inflammatory effects with reduced systematic corticosteroid concentration compared to prednisolone [12]. Therefore, developing a topical delivery system that maximizes Pred MS localization in the mucosal epithelium tissues is highly beneficial for more efficient treatment of RAU.

Nanoparticles provide numerous advantages for buccal drug delivery as they enhance the solubility of hydrophobic drugs and consequently improve their dissolution rates; also, they may provide controlled drug release. Moreover, the nanoparticles can enhance the stability of active ingredients [13]. Furthermore, the utility of cationic nanoparticles in the buccal route has been studied extensively for improving buccal drug delivery as cationic nanocarriers interact with the negatively charged mucosa; thus, they can prolong the drug residence time and consequently enhance drug penetration into the mucosal tissues [14].

Several studies have suggested that buccally applied liposomes provide appropriate drug concentration in the oral mucosal tissues while minimizing systematic drug effects [15,16]. In addition, liposomes can release the encapsulated drug in a more controlled pattern. Moreover, liposomes enhance drug localization in ulcerated areas and enhance anti-inflammatory drugs’ efficacy by targeting mononuclear phagocytic systems [17]. However, encapsulation inefficiencies, aggregation, fusion, partially controlled particle size and poor aqueous stability are the major limitation of liposomes [18,19].

Therefore, several studies have been conducted to prepare novel lipid vesicles with inherent stability. Quatsomes are unilamellar nanovesicles made mainly of cholesterol and quaternary ammonium surfactants (QAS). Quatsomes have excellent stability, homogeneity, and high drug-loading capacity [20]. In addition, quatsomes enhance the bioactivity and stability of proteins. Furthermore, quatsomes can be labeled with fluorescent dyes for bioimaging and biodistribution assays [21]. Therefore, quatsomes are considered promising alternatives to conventional liposomes. Moreover, the presence of QAS that possess antimicrobial activity in the components of quatsomes is considered beneficial in treating wounds [22]. However, the cationic nanoparticles formulated with QAS, especially with dimethyldidodecylammonium bromide (DDAB) or cetyltrimethylammonium bromide (CTAB), bear a greater risk of toxicity, despite several studies that pointed out that the toxicity of QAS has been greatly diminished after association with lipidic molecules as soy lecithin [23,24], further studies are needed to assess the toxicity of quatsomes.

The nanoparticles should be dispersed in a gel base to increase the residence time of the drug on the buccal mucosa. Mucoadhesive gels have a more potent effect, have a longer duration, target specific ulceration sites, and reduce the dosing frequency. Chitosan and polyacrylic acid-based polymers are commonly used gelling agents. Chitosan is a cationic polysaccharide with bioadhesive and antimicrobial properties, and it can extend the contact time of the drug with buccal mucosa [25]. Noveon is a polyacrylic acid-based polymer with good bioadhesive properties due to a high percentage of carboxylic groups that enable hydrogen bonding with mucosal mucin [26].

The initial analyses of the quatsomes found that several independent factors, such as quaternary ammonium surfactants (QAS) and cholesterol (CHO) molar ratio, QAS type, and sonication time, impacted the quatsomes characteristics. Therefore, a 2^3^ factorial design was performed using Design Expert^®^ software version 13 (Stat Ease, Inc., Minneapolis, MN, USA) where the QAS type (X_1_), quaternary ammonium surfactants (QAS) and cholesterol (CHO) molar ratio (X_2_), and sonication time (X_3_) were chosen as independent factors, whereas particle size (PS; Y_1_), polydispersity index (PDI; Y_2_), zeta potential (ZP; Y_3_), entrapment efficiency percent (EE%; Y_4_) and the percentage of drug released after 6 h (Q6%; Y_5_) were set as dependent factors. The optimized quatsomal formula was selected and incorporated into different gel bases to prepare mucoadhesive gel. The mucoadhesive gel was evaluated by determining drug content, rheology properties, spreadability, ex vivo mucoadhesive force, and in vitro drug release. Furthermore, the in vivo performance of the mucoadhesive gel loaded with Pred MS quatsomes in treating RAUs was evaluated via an in vivo pharmacodynamic study, including histological testing and the evaluation of inflammatory biomarkers.

## 2. Materials and Methods

### 2.1. Materials

Prednisolone sodium metazoate (CAS number 630-67-1) was a gift from AL-Andalous Medical Company Ltd. (6th October City, Giza, Egypt). Cholesterol, dimethyldidodecylammonium bromide (DDAB), and cetyltrimethylammonium bromide (CTAB) were obtained from Sigma-Aldrich Chemical Co. (St. Louis, MO, USA). Noveon AA-1^®^ was purchased from Lubrizol Advanced Materials Europe BVBA (Chaussee de Wavre, Brussels, Belgium). Chitosan was purchased from El-Nasr Pharmaceutical Chemicals Co. (Cairo, Egypt). Wistar albino rats were obtained from Misr University for Science and Technology animal center (6th October City, Giza, Egypt).

### 2.2. Preparation of Pred MS-Loaded Quatsomes

Quatsomes were prepared with varied QAS: CHO molar ratios (Table 1), where 20 mg of Pred MS was added to weighed amounts of CHO and QAS (DDAB or CTAB), then 10 mL of Milli-Q water was added, so the final drug concentration within formula was 0.2% *w/v*, then the mixture was sonicated for the required time at 30/10 s on/off cycle and 40% amplitude using a QSonica probe sonicator (Newton, CT, USA) after being placed into an ice bath. Then, the dispersion was stirred overnight at room temperature using a magnetic stirrer (LX653DMS, LabDex, London, UK) at 400 rpm [27]. Finally, the quatsomal formulae were stored in the refrigerator at 4 °C until characterization.

### 2.3. Characterization of Pred MS-Loaded Quatsomes

#### 2.3.1. Determination of PS, PDI, and ZP of Quatsomes 

The PS, PDI, and ZP of quatsomes formulae were measured using the Malvern Zetasizer (Malvern Instruments Ltd., Malvern, UK). The assessments were performed after a 10-fold dilution with Milli-Q water [28].

#### 2.3.2. Determination of EE%

Pred MS entrapped in quatsomes was identified by using a dialysis approach. Briefly, 1 mL of the sample was loaded in semi-permeable membrane tubing (MW of 12,000–14,000 Dalton), immersed in 50 mL of distilled water, and agitated at 60 rpm for 1 h using a magnetic stirrer. The free Pred MS was dialyzed until consistent concentration was achieved. The free amount of Pred MS was determined spectrophotometrically (Shimadzu UV-1601 PC; Kyoto, Japan) at λmax 237 nm. The EE% was calculated using the following equation [29,30].
(1)$$EE\left(\%\right)=\frac{Total\;Pred\;MS\;amount-Diffused\;Pred\;MS\;amount}{Total\;Pred\;MS\;amount}\times 100$$

#### 2.3.3. In Vitro Drug Release Study

The in vitro release of Pred MS quatsomes was assessed using a USP dissolution apparatus II (model). First, the cellulose membrane (12,000–14,000 Mwt cutoff) was soaked in phosphate buffer saline (PBS) solution of pH 6.8 overnight at room temperature. Next, 1 mL of the formula (2000 µg of Pred MS) was placed in a plastic cylindrical tube with one end tightly covered with a cellulose membrane and the other attached to the USP dissolution apparatus shaft instead of the baskets. The release media, 50 mL PBS (pH 6.8) applied in the apparatus baskets, was continuously stirred at 50 rpm with the paddle at 37 °C [31]. A 1 mL sample was collected at definite time intervals, and the quantity of Pred MS released was measured spectrophotometrically at 237 nm using a UV spectrophotometer (UV-1601, Shimadzu, Japan). The withdrawn samples were replaced with an equivalent volume of fresh media to ensure sink conditions. The samples were drawn at 0.25, 0.5, 1, 2, 3, 4, 5, and 6 h. The percent of drug released after 6 h (Q6%) was determined. To determine the drug release mechanism of Pred MS from different quatsomal formulae, the in vitro release profiles were fitted to various kinetic models.

### 2.4. Optimization of Pred MS-Loaded Quatsome

A 2^3^ factorial design experiment was conducted where the independent variables were the QAS type (X_1_), QAS: CHO molar ratio (X_2_), and sonication time (X_3_) at two levels in the design of the formulation, while the dependent variables were the particle size (PS: Y_1_), polydispersity index (PDI; Y_2_), zeta potential (ZP; Y_3_), entrapment efficiency percent (EE%; Y_4_) and drug released after 6 h (Q6%; Y_5_).

### 2.5. Selecting the Optimized Pred MS-Loaded Quatsomes Formula

The desirability tool was used to choose the optimized formula. The selection was decided to obtain a quatsomal formula with the least PS and PDI and the highest Q6%, EE%, and ZP values.

### 2.6. Morphology

The shape of the optimized quatsomal formula was evaluated via a transmission electron microscope (TEM) (JEM-1230, Joel, Tokyo, Japan). First, 50 µL of freshly prepared quatsomal formula was dropped on a carbon-coated grid, 2% phosphotungstic acid was added for negative staining, and then dried for visualization [32].

### 2.7. Compatibility Evaluation of the Optimized Quatsomal Formula Using Fourier Transform Infrared (FTIR)

FTIR spectroscopy revealed possible interactions between Pred MS and the ingredients of the optimized quatsomal formula (CHO, QAS). The spectra were obtained for pure Pred MS, DDAB, CHO, and the optimized lyophilized formula using FTIR 8400 (Shimadzu, Kyoto, Japan), where almost 2 mg of each sample were blended with potassium bromide and scanned between 400 and 400 cm^−1^ [33].

### 2.8. Stability Study of the Optimized Quatsomal Formula

The optimized quatsomal formula (containing both encapsulated and unencapsulated Pred MS) was stored at 4 °C and 25 °C for 3 months and inspected regarding the EE%, PS, PDI, and ZP. The data were compared with those of the fresh quatsomal formula. Besides, the system was visually observed for sedimentation or any signs of particle aggregation. The significance of the outcomes of the parameters under examination was examined via *t*-test. The analyses were conducted in triplicate [34,35].

### 2.9. Formulation of Pred MS Mucoadhesive Quatsomal Gel

The optimized formula was incorporated into different gel bases using Noveon or chitosan polymers in different concentrations (1 or 3%) to prepare a mucoadhesive gel. To formulate the chitosan gel base, 0.5% of glacial acetic acid was added to half the required water (10 mL); then, chitosan was added and stirred slowly. After the swelling, the remaining 10 mL of water was dropped and mixed. The gels were kept at room temperature overnight to remove the air bubbles. To formulate the Noveon gel bases, polymeric solutions of Noveon were prepared by gradual dispersion of the required weights in deionized water and stirring at 900 rpm for 30 min. The pH of the solution was adjusted to 6.8 ± 0.2 (physiological buccal mucosa pH) with triethanolamine. The purified quatsomal pellets encapsulating the required amount of Pred MS were added to the blank gel bases with continuous mixing using a magnetic stirrer to reach a final concentration of Pred MS 0.2% *w/w* [31,36].

### 2.10. Evaluation of Pred MS Mucoadhesive Quatsomal Gel

#### 2.10.1. Evaluation of the Physical Properties

Visual inspection observed quatsomal gels for physical appearance, consistency, and color.

#### 2.10.2. Determination of Gel pH

The pH of different gel formulations was determined using a digital pH meter (Jenway TM 3510, Staffordshire, London, UK). A magnetic stirrer dispersed one gram of each quatsomal gel in 9 mL of distilled water. Then, the pH of the buccal gels was estimated [37].

#### 2.10.3. Determination of Drug Content

Briefly, one gram of each quatsomal gel was magnetically dispersed in 100 mL methanol to extract the drug, then stirred for 2 h. Finally, the solution was filtered and analyzed spectrophotometrically at 237 nm [38].

#### 2.10.4. Evaluation of the Rheological Properties

A sample of each prepared formula was placed in the adaptor of the Brookfield digital DV-III Model viscometer. At room temperature, the rotating speed progressively rose from 25 to 250 rpm [39]. Spindle CP-52 and 25 rpm were used for measuring the viscosity.

#### 2.10.5. Assessment of Gel Spreadability

Briefly, the gel was prepared 48 h before the test, and one gram sample of each gel was sandwiched between two slides by adding a specific weight of 20 g for 60 s. Then the diameter of the sample between the plates was measured. Finally, the spreadability was calculated by the following equation [40]:(2)$$S_{i}=M\times \left(\frac{L}{T}\right)$$

Si is the spreading area, M is 20 g, L is the length of a glass slide (9.8 cm), and T is 20 s.

#### 2.10.6. In Vitro Drug Release Study

The in vitro release was performed using 1 gm of the quatsomal gel, as mentioned above in Section 2.3.3.

#### 2.10.7. Ex Vivo Mucoadhesive Force (MF)

Utilizing a modified two arms physical balance technique, freshly isolated buccal mucosa from rabbits was used to determine the mucoadhesive force (MF). On the same day of isolation, buccal mucosa was harvested from a nearby abattoir and utilized. In this technique, a base with a slide replaced one of the balance arms where the mucosa was affixed to the slides fronting the mucoadhesive gels. The gel was then put on the slide and covered with another moving slide attached to the balance’s arm via wire. When the whole system was motionless, the weight on the left pan was increased by adding water drops to create tension on the wire. The MF value was estimated based on the strength that made the gel detach from either mucosa surface [41]. The MF values of different quatsomal gels were estimated from the following equation MF.
MF (dyne/cm^2^) = (m × g)/A(3)
where (A) is the buccal mucosa area in cm^2^, (m) is the weight of water in g, and (g) is the acceleration due to gravity taken as 981 cm/s^2^.

### 2.11. In Vivo Study

The protocol was approved by the Research Ethics Committee of the Faculty of Pharmacy, Cairo University (Approval no. PI 2353), following the “Guide for the Care and Use of Laboratory Animals” stated via the Institute of Laboratory Animal Research (Washington, DC, USA).

Twenty-four Wistar albino rats (weighing about 200 g each) were classified into four groups of 6 animals each; the first group (GP 1) was set as the negative control, while the other three groups were exposed to ulceration of the buccal mucosa where a round filter paper, 6 mm in diameter, was soaked in 15 μL of 50% acetic acid and was used to cause aseptic tissue necrosis [42]. The acid-soaked paper was applied to the rat’s labial gingival tissue for 60 s to induce circular ulcers. Group (GP 2) was set as positive control while the third group (GP 3) was treated with a gel containing 0.2% *w/w* Pred MS, and the fourth group (GP4) was treated with Pred MS mucoadhesive quatsomal gel. The treatment of GP3 and GP4 was twice a day for 8 days. The wound healing was evaluated by measuring the reduction in the ulcer area.

At the end of the experiment (day 8), a histopathological study was conducted where three animals from each group were sacrificed with an intramuscular injection of ketamine (30 mg/kg). First, the rat buccal mucosa was separated and fixed in 10% formalin in phosphate buffer and then embedded in paraffin blocks. Next, the tissues were stained with hematoxylin-eosin. All sections were digitally photographed, and the images were analyzed using a computer-assisted image analyzer system of an Olympus BX-51 microscope (Tokyo, Japan) outfitted with an Olympus DP-71 high-resolution video camera. In addition, on day 8, another three rats from each group were given anesthesia and blood samples were taken to assess high-sensitivity C-reactive protein (hs-CRP), interleukin-6, and tumor necrosis factor-alpha (TNF-alpha) using enzyme-linked immunosorbent assay (ELISA) according to the manufacturer’s instructions [43].

### 2.12. Statistical Analysis of Data 

The one-way analysis of variance (ANOVA) test was used to evaluate whether there was a significant difference between the study outcomes. The significance level was set at 0.05, and (* *p* < 0.05) was considered statistically significant.

## 3. Results and Discussion

### 3.1. Factorial Design Optimization

In the initial analysis of the factors that might affect the properties of the quatsomal formulae, it was found that QAS and CHO molar ratio, QAS type, and sonication time impacted the quatsomes characteristics. Therefore, the QAS type (X_1_), QAS to CHO molar ratio (X_2_), and sonication time (X_3_) were selected as independent factors for the design, as seen in Table 2 and Table 3.

### 3.2. Effect of Formulation Variables on the Particle Size

The PS of quatsomes varied from 69.47 ± 0.41 to 113.28 ± 0.79 nm, as displayed in Figure 1a,b. These results conform to some extent with prior studies that observed that the diameter of quatsomes ranged from 50 to 100 nm [28]. ANOVA results specified that all the dependent variables significantly affected the PS of quatsomes. Regarding (X_1_), smaller vesicles were obtained by CTAB compared to those formulated by DDAB, as CTAB possesses a smaller molecular volume than DDAB [44]. Increasing the molar ratio of QAS to CHO (X_2_) from 1:1 to 1:2 led to a reduction in the quatsomes size; these results agree to some extent with Hassan et al. 2020 who found that the PS of quatsomes was significantly dependent on QAS to CHO molar ratio where increasing the QAS molar ratio led to increasing PS of quatsomes as the QAS molecules aggregated to micelles rather than interacting with CHO [45]. In this study, results can be inferred as increasing CHO molar ratio might decrease the aggregation of QAS molecules into micelles by increasing their hydrophobic interaction. The sonication time (X_3_) had a negative effect on the quatsomes size due to the breaking of the vesicles [46].

The regression equation was
PS = +90.62546875 + 3.77046875 ∗ A − 5.19453125 ∗ B − 10.86578125 ∗ C − 1.36453125 ∗ AB − 4.45328125 ∗ AC − 2.49078125 ∗ BC − 0.84828125 ∗ ABC.

### 3.3. Effect of Formulation Variables on Polydispersity Index

The PDI of quatsomal formulae was less than 0.35, as displayed in Figure 2a,b; these results comply with the prior studies that point out the homogeneity of the quatsomes [47]. The increased surface charge of vesicles reduces aggregation and fusion, leading to more homogenous dispersion and reduced PDI value. Accordingly, the DDAB-based formulae showed the lowest PDI values; due to the higher surface charge obtained by DDAB formulae. The superiority of DDAB over CTAB in terms of quatsomes homogeneity may be explained by the fact that DDAB has higher steric stabilization and stronger intermolecular interactions than CTAB [48]. In addition, increasing the molar ratio of CHO (X_2_) showed a significant (*p* < 0.0001) reduction in the PDI values due to its stabilizing effect that reduces the fusion and aggregation of vesicles [49]. Increasing sonication time led to the formation of more homogenous vesicles because the prolonging of the ultrasound streaming provides higher mixing and hence increases the homogeneity of the quatsomes [50]. The regression equation was
PDI = +0.266125 − 0.00375 ∗ A − 0.035625 ∗ B − 0.02275C − 0.009 ∗ AB + 0.007875 ∗ AC − 0.0005 ∗ BC + 0.001625 ∗ ABC.

### 3.4. Effect of Formulation Variables on Zeta Potential

The values of ZP of quatsomal formulae ranged from 45.15 ± 0.19 to 68.1 ± 0.28 mV, as shown in Figure 3a,b; one of the extraordinary advantages of quatsomes is their excellent colloidal stability, leading to the formulation of nano drug delivery systems with elevated quality [24]. The type of QAS (X_1_) significantly affected the ZP values of the formulae (*p* = 0.0004), where DDAB-based formulae revealed greater ZP values. Although both QAS molecules carry one positive charge, which would result in similar charge densities, the configuration of the different QAS molecules within the vesicular membrane might increase the exposure of the positive charge of DDAB compared to CTAB, thus revealing higher ZP. A similar finding was observed by Refai et al. (2022) [51]. Regarding the effect of increasing the CHO molar ratio (X_2_) on the ZP values of the quatsomal formulae, it was found that the ZP values increased by increasing the CHO molar ratio; these results might be due to the negative effect of CHO molar ratio on quatsomes PS where increasing CHO molar ratio led to a significant reduction in quatsomes PS which led to increasing surface ratio and subsequently increased ZP values. These findings agree with Cano-Sarabia et al. (2010), who found that decreasing CHO content of the vesicular system composed of CHO and CTAB resulted in increased PS with a subsequent decrease in ZP [52]. In addition, increasing the sonication time (X_3_) led to a significant increase in the ZP values, as diminishing the size of the vesicles led to an increase in the surface ratio [53]. The regression equation was
ZP = +56.743125 + 3.334375 ∗ A + 1.138125 ∗ B + 5.649375 ∗ C + 0.146875 ∗ AB − 0.884375 ∗ AC + 1.244375 ∗ BC + 1.078125 ∗ ABC.

### 3.5. Effect of Formulation Variables on Entrapment Efficiency%

The EE% of quatsomal formulae ranged from 98.60 ± 1.22 to 79.62 ± 1.44%, as displayed in Figure 4a,b, where the type of QAS (X_1_) significantly affected the EE % of the formulae as DDAB with the higher lipophilicity (log *p* = 11.8) helped in entrapping Pred MS (log *p* = 2.8) to a greater extent than CTAB of lower lipophilicity (log *p* = 8) [54]. On the other hand, increasing the molar ratio of CHO (X_2_) led to a significant reduction in the EE%. This finding might be due to the association of the CHO molecule to the hydrocarbon chain of the QAS due to its hydrophobic character. However, the mismatch in size between the molecules of the QAS and CHO with the increased rigidity of the CHO molecule probably produced a deformation of the polar ammonium headgroup of the QAS around the polar oxygen of the hydroxyl group of CHO. This deformation of the cationic ammonium head group of QAS might have affected its electrostatic attraction with the negatively charged drug (Pred MS) and subsequently affected the EE% [55]. Increasing the sonication time (X_3_) led to a significant decrease in the EE%, where the prolonged sonication time caused large vesicles to be reformulated to smaller-sized vesicles, leading to drug leakage from quatsomes [56]. The regression equation was
EE% = +91.1659375 + 5.582625 ∗ A − 2.5965 ∗ B − 1.134 ∗ C + 1.0496875 ∗ AB + 0.6149375 ∗ AC − 0.3590625 ∗ BC + 0.339375 ∗ ABC.

### 3.6. Effect of Formulation Variables on the In Vitro Drug Release

The cumulative drug release percentage at different time intervals is illustrated in Figure 5. The Q6% of different quatsomal formulae varied from 58.39 ± 1.75 to 94.42 ± 2.15%. It was noticed from Figure 6a,b, that the inclusion of the DDAB into quatsomal formulae enhanced the in vitro drug release more than CTAB (*p* < 0.0001). This finding could be attributed to the higher flexibility of the double chained DDAB, compared to the single chained CTAB, resulting in enhanced drug release from the quatsomal formulae [48]. On analyzing the in vitro release results, F4 and F7 revealed the highest Q6% (93.89 ± 1.19 and 94.42 ± 2.15, respectively) compared to the other quatsomal formulae. Both formulae comprised the same QAS to CHO molar ratio (1:2) but were prepared at different sonication times, 10 and 20 min, respectively. Contrary to previous studies, which confirmed that increasing CHO in the self-assembled nanovesicles led to an increase in the rigidity of the vesicles and hence reduced the drug in vitro release, in our study, increasing CHO molar ratio (X_2_) led to an increase in the Q6%. At pH 6.8, the sulphonate group in Pred MS becomes ionized, negatively charged, which would enhance its aqueous solubility. In addition, the amino group of QAS at the same pH becomes protonated, positively charged, resulting probably in the rearrangement of the QAS in the nanosystem [45]. We assume that increasing the CHO led to an increase in the hydrophobicity of the quatsomal core and consequently increased the CHO- water repulsion, thus with simultaneous rearrangement of the QAS in the quatsomes and the enhanced solubility of Pred MS at pH 6.8, the release of Pred MS was augmented. In addition, prolonging the sonication time (X_3_) revealed a positive influence on the Q6%; which might be attributed to their negative impact on the vesicle size as the reduction in the vesicle size increased the area exposed to the release media and reduced the path length for of drug diffusion and thus increasing the Q6%. The regression equation was
Q6% = +77.815 + 11.65 ∗ A + 3.35375 ∗ B + 2.9125 ∗ C + 1.17375 ∗ AB − 2.765 ∗ AC − 0.39875 ∗ BC 0.0712 ∗ ABC. 

The regression coefficient values recommended that the in vitro release profile of Pred MS from all quatsomal formulae could best be fitted using the Higuchi release model.

### 3.7. Selection of the Optimized Formula

The optimized formula (desirability = 0.992) that met the desired criteria comprised DDAB as QAS, 1:2 QAS to CHO molar ratio with a sonication time of 10 min (F4).

### 3.8. Morphology

The TEM micrograph of the optimized quatsomal formula (Figure 7) shows that the vesicles were distinct and spherical. In addition, the PS measured by Zetasizer agrees with TEM results.

It is worth noting that the TEM micrographs proved the successful formation of quatsomes at a QAS: CHO ratio of 1:2. This finding disagrees with preceding studies, which showed that quatsomes were only formulated at an equimolar proportion of QAS and CHO. Ferrer-Tasies et al. (2013) postulated that the formation of quatsomes is a synergy between one sterol and one single-tail QAS (CTAB) in an equimolar proportion, which allows the self-assembly into multicomponent membrane vesicles in aqueous media and on increasing the CHO molar ratio CHO crystals would exist with the formation of elongated and distorted vesicles [55]. In contrast, the TEM micrographs in this study showed no distorted vesicles but rather spherical ones without CHO crystals. This could be attributed to the fact that double-tail QAS (DDAB) could synergize with two molecules of CHO, not only one as in CTAB formulae, allowing the successful formation of the quatsomes.

### 3.9. Compatibility Assessment of the Optimized Formula Using Fourier Transform Infrared (FTIR)

Pure Pred MS FTIR spectrum (Figure 8a) displayed peaks at 3444.87 cm^−1^ and 3421.72 cm^−1^, characteristic for O-H stretching, indicative of the presence of the hydroxyl groups and peaks at 1714.72 cm^−1^,1653.0 cm^−1^ and 1610.56 cm^−1^ for the existence of the three carbonyl groups as well as a sharp peak at 1269.16 cm^−1^ characteristic for S=O stretching. On the other hand, the CHO FTIR spectrum (Figure 8b) displayed a peak at 3444.87 cm^−1^, characteristic for O-H stretching, a peak at 2933.73 cm^−1^ due to asymmetric and symmetric stretching vibration of methylene and methyl groups, and a peak at 1464 cm^−1^ indicative of methylene and methyl groups deformation vibrations. Regarding the lyophilized formula’s FTIR spectrum (Figure 8c), the disappearance of a peak at 1610.56 cm^−1^ characteristic for carbonyl group stretching, as well as a C-N bending vibration peak at 1487.12 cm^−1^ could be noticed, in addition, a weak peak of at 1269.16 cm^−1^ characteristic for S=O stretching could be detected. This finding indicates the interaction between the negatively charged sulphonyl group of the drug and the positively charged ammonium group of the QAS. The DDAB (Figure 8d) showed peaks at 2920.23, 2848.86, 1463.97, and 719.45 cm^−1^ for methylene stretching vibration, 2943.37 cm^−1^ for methyl stretching vibration and a sharp peak at 1487.12 cm^−1^ for bending vibration of C-N in the ammonium group. The chemical structures of Pred MS and the components of the optimized quatsomal formula are shown in Figure 9.

### 3.10. Stability Study of the Optimized Quatsomal Formula

The quatsomal formula showed no sedimentation or vesicle aggregation during the storage time. The EE%, PS, PDI, and ZP measurements were 93.20 ± 1.62%, 105.24 ± 2.34 nm, 0.243 ± 0.009, and 70.4 ± 1.42 mV after storage at 4 °C and 94.33 ± 1.55%, 104.36 ± 1.95 nm, 0.255 ± 0.008, and 67.3 ± 1.49 mV after storage at 25 °C. These results showed insignificant variation from the freshly prepared quatsomes (paired *t*-test, *p* > 0.05), indicating the quatsomal formulae’s physical stability. This finding complies with several studies that assured the excellent stability of quatsomes [58].

### 3.11. Evaluation of Pred MS Mucoadhesive Quatsomal Gel

#### 3.11.1. Physical Appearance of the Gel

All tested gels were clear, homogenous, and free from air bubbles.

#### 3.11.2. Determination of Gel pH

The pH of the quatsomal gels ranged from 5.8 ± 0.13 to 6.9 ± 0.1, as shown in Table 4, which is close to the buccal pH (6.8) and will be, therefore, non-irritant to the oral mucosa [59]. It is worth noting that Noveon gels have a more compatible pH with the buccal mucosa than chitosan gels.

#### 3.11.3. Determination of Drug Content

To ensure that the purified quatsomes pellets were uniformly dispersed within the gel, the different quatsomal gel formulae were tested for the actual drug content, which was compared with the theoretical drug content (0.2% *w*/*w*). The results showed that the actual drug content of the gel formulae ranged from 94.9 ± 0.45 to 97.3 ± 0.92%, within the desired range of 90 to 110% suggesting that the quatsomal formula was uniformly dispersed within different gel bases.

#### 3.11.4. Evaluation of the Rheological Properties

All the quatsomal gel formulae showed a pseudoplastic flow behavior, presenting an immediate flow after stress application. In our study, four quatsomal gels were formulated using mucoadhesive polymers (chitosan and Noveon) in 1 and 3% concentrations. The viscosity of the gels was proportional to the concentration of the polymers (*p* < 0.0001); therefore, chitosan and Noveon formulae at 3% polymer concentration (G2 and G3) showed higher apparent viscosity of 9176 ± 59.75 cp and 7938 ± 45.76 cp, respectively compared with 1% gel formulations (Table 4). A high viscosity value is required for the mucoadhesive gel as it leads to longer residence time in the buccal mucosa and consequently enhances the effectiveness of the quatsomal formula. Furthermore, a more viscous formula has the advantage of a slow flow index, which minimizes possible side effects of accidental swallowing [60].

#### 3.11.5. Assessment of Gel Spreadability

The spreadability values ranged from 5.5 ± 0.75 to 12.91 ± 0.59 (g.cm/s), which indicates that all the formulated gels will spread easily at low shear stress. The spreadability values depended on the gelling agent concentration, where increasing polymer concentration significantly (*p* < 0.0001) minimized the values [61], as the viscosity is inversely proportional to the spreadability [62]. Furthermore, the chitosan-based formulae showed significantly higher values than the Noveon-based formulae.

#### 3.11.6. In Vitro Drug Release Study

The % of the Pred MS released from quatsomal gel mainly depended on the polymer concentration (*p* < 0.0001), as displayed in Figure 10, where the Q6% of Pred MS from gels containing 3% polymer concentration was lower as the polymeric network became more entangled [63]. Moreover, the gel formula comprised higher polymer % uptake of less water; therefore, the dissolution and erosion rates were reduced. Furthermore, drug diffusion is negatively impacted by the volume fraction occupied by the polymeric agent, according to Laufferʼs polymer gel molecular diffusion theory [64].

#### 3.11.7. Ex Vivo Muco-Adhesive Force

The retention time of the gel formulae at the buccal mucosa mainly depends on its mucoadhesion properties. A greater MF would increase the retention time and hence the effectiveness of the formulations [65]. Many studies stated that chitosan-based gels exhibited better mucoadhesion properties than polyacrylic acid gels due to the electrostatic attraction of the positive amino groups of chitosan with the negatively charged mucus [66]. However, in our study, quatsomal gel prepared with Noveon showed higher mucoadhesion than chitosan-based gel. This finding agrees with Cevher et al. (2008), who found that polyacrylic acid-based gels showed higher mucoadhesive properties than chitosan-based gels. The author suggested that the nonionized carboxylic acid groups found in the Noveon bind directly to the polysaccharide groups of mucin by hydrogen bonds, showing increased mucoadhesion [67,68]. In addition, the polymer concentration significantly impacted the mucoadhesion of the prepared quatsomal gels, as increasing the polymer concentration would increase the number of penetrating polymer chains per unit volume of mucin. Accordingly, the polymer and mucin interaction would probably become more stable, increasing the gel’s MF.

Based on all previous investigations, the quatsomal mucoadhesive gel (G2) was selected for the in vivo assessment as it showed the highest MF and acceptable Q6%.

### 3.12. In Vivo Study

In this study, the wound-healing potential of the developed Pred MS mucoadhesive quatsomal gel was investigated in rats with induced buccal ulcers. This study was done in comparison to the same gel base containing an equivalent amount of the drug. On day 4, there was a significant decrease in the ulcer size in the quatsomal gel-treated group (GP4) compared to the positive group (GP2) and the free drug gel-treated group (GP3); moreover, there was no significant difference between the positive group (GP2) and free drug gel treated group (GP3). On day 8, there was a significant reduction in the ulcer size in all groups compared to day 4; however, the superiority of the quatsomal gel in treating the aphthous ulcers was evident compared to free drug gel, *p* < 0.05, as shown in Figure 11, Figure 12 and Figure 13.

The histopathological examination showed similar results where on day 4, the negative control (GP1) demonstrated the standard structure of oral mucosa marked in its intact layers as displayed in Figure 13a,b, where the epidermis exhibited normal lining of the stratified squamous non-keratinized epithelium (arrows). In addition, the dermis revealed intact dense irregular connective tissue (wave arrows) supported with blood vessels (arrowhead) as well as skeletal muscle fibers (star). At the same time, the positive control (GP2) (Figure 13c,d) showed complete loss of the epidermal layer and the existence of a thick layer homing ulcerated necrotic tissue (arrows) mixed with fibrin of clot formation (wave arrows). Moreover, an interstitial hemorrhage (curvy arrows) was also noticed, besides infiltrating a massive number of mononuclear inflammatory cells (thick arrow). The results of the free drug gel-treated group (GP3) (Figure 13e,f) showed emphasized loss of epidermal layer with the development of ulceration and necrotic tissue (arrow) in a moderate grade, hemorrhage (curvy arrow), aggregated mononuclear inflammatory cells (arrowheads), along with edema (cube) leading to dispersion between fibrous connective tissue (wave arrow). The dermal layer displayed granulation tissue that underscored high-fat cells, stromal cells, mononuclear inflammatory cells (star), and unorganized collagen fibers (circle). The results of the quatsomal gel-treated group (GP4) (Figure 13g,h) showed an epidermal layer with the growth of a limited layer of necrotic tissue (arrows) and the existence of interstitial hemorrhage (arrow with tail). In addition, some mononuclear inflammatory cells could be noticed along the epidermis and dermis layers (arrowheads). The dermis marked the granulation tissue homing fat cells (wave arrow).

On day 8, the negative group (GP1) showed the same results as on day 4, while the positive group (GP2) showed exposed scar development in the superficial layer (arrows). Beneath it, granulation tissue (circle) emphasized interstitial hemorrhage (arrow with tail), newly formed blood vessels (wave arrows), and mononuclear inflammatory cells (arrowheads), in addition to limited areas with vacuolation and edema leading to dispersion between fibrous connective tissue (curvy arrow) as displayed in Figure 13i,j. At the same time, the free drug gel-treated group (GP3) showed marked hypertrophic scar along with migrating epidermal cells and invaginated Reta ridges formation in the epidermal layer (arrows). The dermis is detected in limited areas with well-developed fibrous connective tissue (circle) and the other regions characterized by the existence of dilated blood vessels (wave arrows), few inflammatory cells (arrowheads) as well as edema leading to dispersion between fibrous connective tissue (curvy arrows) as illustrated in Figure 13k,l). In contrast, the results of the quatsomal gel treated group (GP4) (Figure 13m,n) showed some migrating epidermal cells and invaginated Reta ridges in the epidermal layer (arrows). In addition, subepidermal edema was observed in limited areas (curvy arrow). The dermis exposed the best progress, evidenced by well-organized fibrous connective tissue (circle), a high number of newly formed blood vessels (wave arrows), in addition to few inflammatory cells (arrowheads).

The results of inflammatory markers (CRP, IL-6, and TNF-alpha) showed that the free drug gel (GP3) caused a partial reduction in their levels compared to the positive control (GP2) (*p* < 0.05) however failed to restore the levels to normal levels (*p* < 0.05) when compared to the negative group (GP1) as displayed in Figure 14. On the contrary, the quatsomal gel (GP4) succeeded in restoring the levels of the inflammatory markers to normal levels when compared to the negative group (GP1) (*p* > 0.05).

These results assured the effectiveness of the quatsomal gel when compared to the free drug gel. The superiority of the quatsomal gel in healing the ulcers might be due to the positive charge of the quatsomes providing close contact between the diseased mucosa and the drug for a prolonged time while releasing the drug due to the electrostatic attraction with the negatively charged mucus. Furthermore, the presence of the QAS in the components of quatsomes gained the formulae antibacterial activity that might have assessed the rapid healing of the ulcers.

## 4. Conclusions

The results obtained in the present work reveal that quatsomes are a promising alternative to conventional liposomes and other lipid-based vesicles. The CHO: QAS molar ratio, the type of QAS and the sonication time impacted the characteristics of the prepared quatsomes. The quatsomal mucoadhesive gel induced almost complete restoration of the normal histology of the ulcerated areas in the studied animals. Conclusively, quatsomal mucoadhesive gel provided promising platforms for the topical delivery of Pred MS in managing recurrent aphthous ulcers. However, further studies are required to assess the toxicity of quatsomes.

## Figures and Tables

**Figure 1 pharmaceutics-15-01947-f001:**
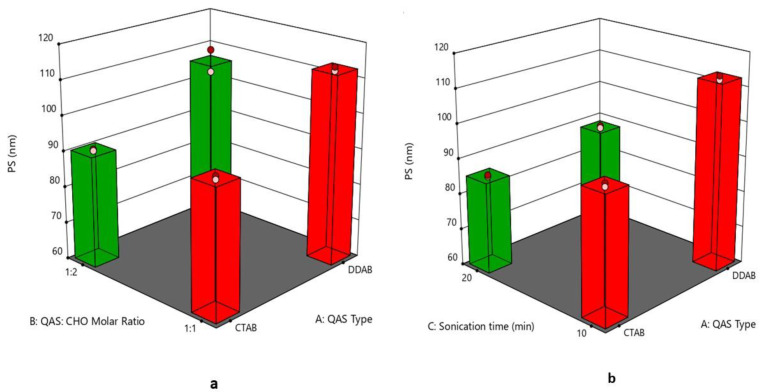
(**a**) 3D plot for the effect of (QAS) type (A), (QAS), and (CHO) molar ratio (B) on PS, (**b**) 3D plot for the effect of (QAS) type (A) and sonication time (C) on PS. Abbreviation: QAS—quaternary ammonium surfactants; CHO—cholesterol; PS—particle size.

**Figure 2 pharmaceutics-15-01947-f002:**
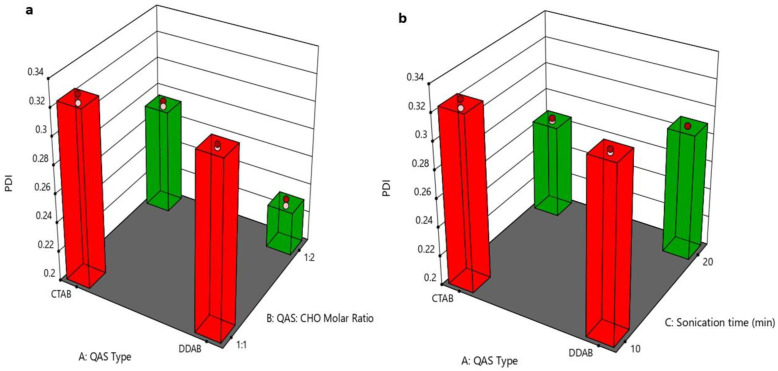
(**a**) 3D plot for the effect of (QAS) type (A), (QAS), and (CHO) molar ratio (B) on PDI, (**b**) 3D plot for the effect of (QAS) type (A) and sonication time (C) on PDI. Abbreviation: QAS—quaternary ammonium surfactants; CHO—cholesterol; PDI—polydispersity index.

**Figure 3 pharmaceutics-15-01947-f003:**
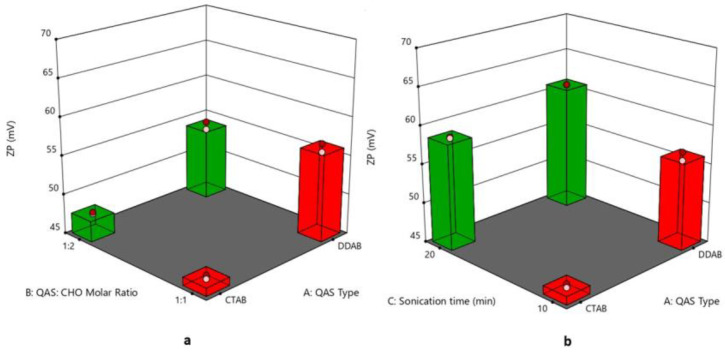
(**a**) 3D plot for the effect of (QAS) type (A), (QAS), and (CHO) molar ratio (B) on ZP, (**b**) 3D plot for the effect of (QAS) type (A) and sonication time (C) on ZP. Abbreviation: QAS—quaternary ammonium surfactants; CHO—cholesterol; ZP—zeta potential.

**Figure 4 pharmaceutics-15-01947-f004:**
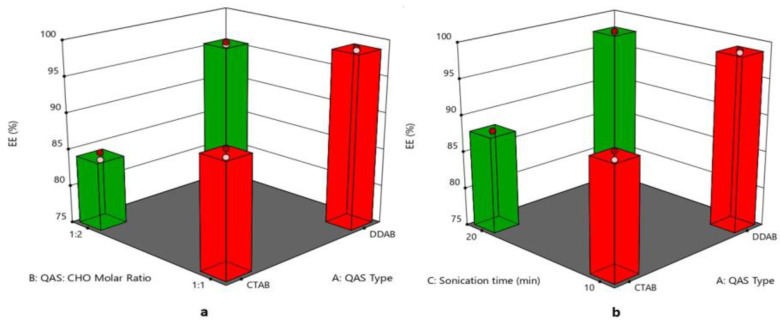
(**a**) 3D plot for the effect of (QAS) type (A), (QAS), and (CHO) molar ratio (B) on EE, (**b**) 3D plot for the effect of (QAS) type (A) and sonication time (C) on EE. Abbreviation: QAS—quaternary ammonium surfactants; CHO—cholesterol; EE—entrapment efficiency.

**Figure 5 pharmaceutics-15-01947-f005:**
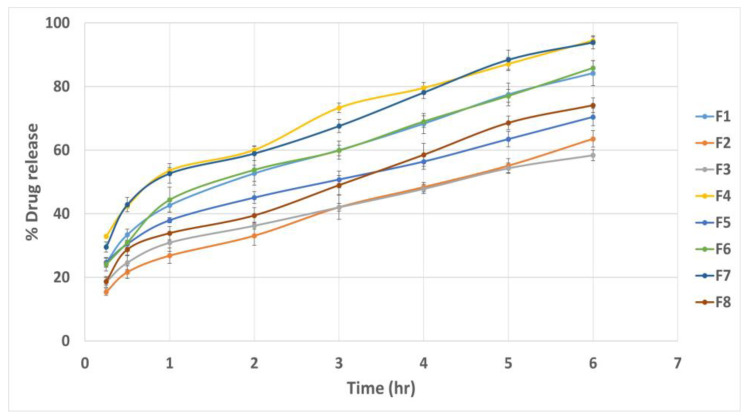
In vitro drug release profiles from different quatsomal formulae.

**Figure 6 pharmaceutics-15-01947-f006:**
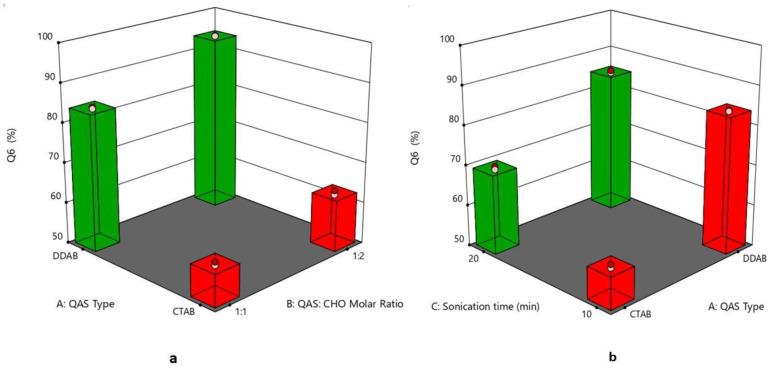
(**a**) 3D plot for the effect of (QAS) type (A), (QAS), and (CHO) molar ratio (B) on Q6%, (**b**) 3D plot for the effect of (QAS) type (A) and sonication time (C) on Q6%. Abbreviation: QAS—quaternary ammonium surfactants; CHO—cholesterol; Q6%—percent of drug released after 6 h.

**Figure 7 pharmaceutics-15-01947-f007:**
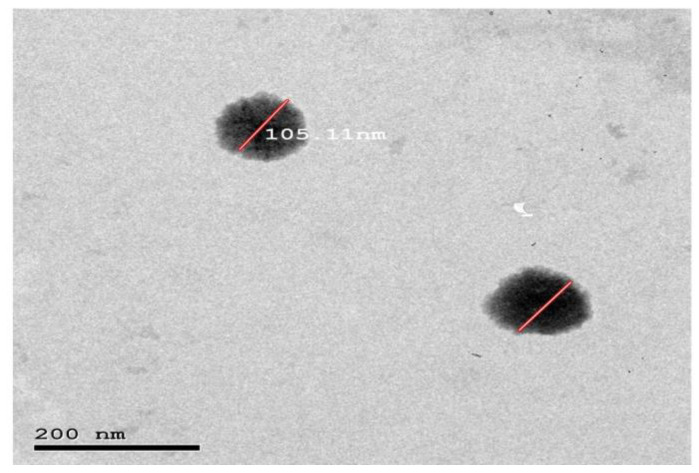
Morphology of the optimized quatsomal formula (F4).

**Figure 8 pharmaceutics-15-01947-f008:**
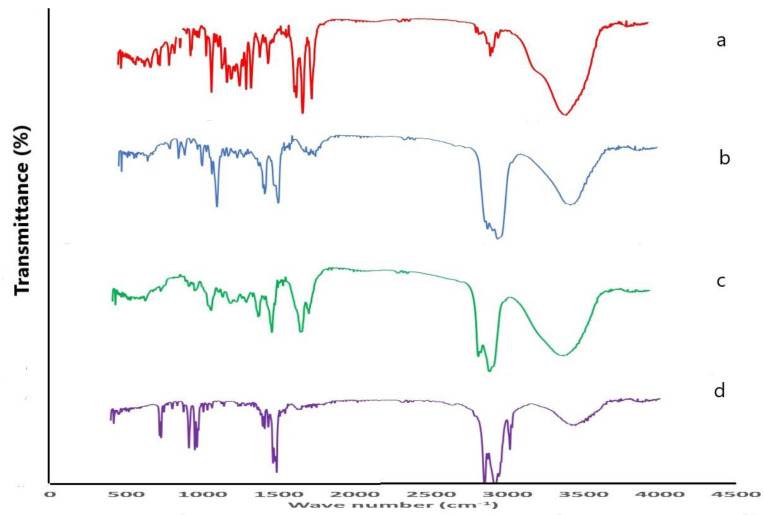
FTIR spectrum of (**a**) prednisolone sodium metazoate, (**b**) cholesterol, (**c**) lyophilized quatsomal formula, (**d**) dimethyldidodecylammonium bromide.

**Figure 9 pharmaceutics-15-01947-f009:**
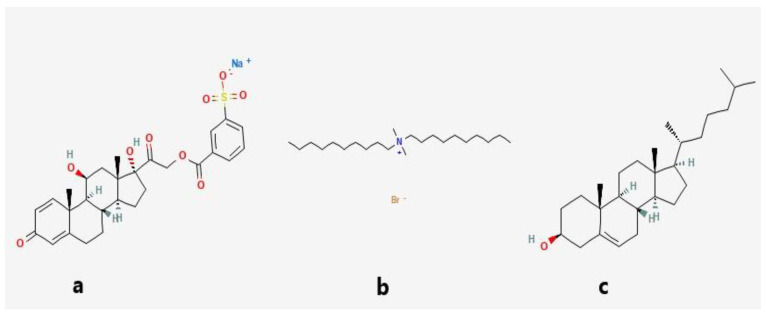
Chemical structure of (**a**) prednisolone sodium metazoate, (**b**) dimethyldidodecylammonium bromide, and (**c**) cholesterol [57].

**Figure 10 pharmaceutics-15-01947-f010:**
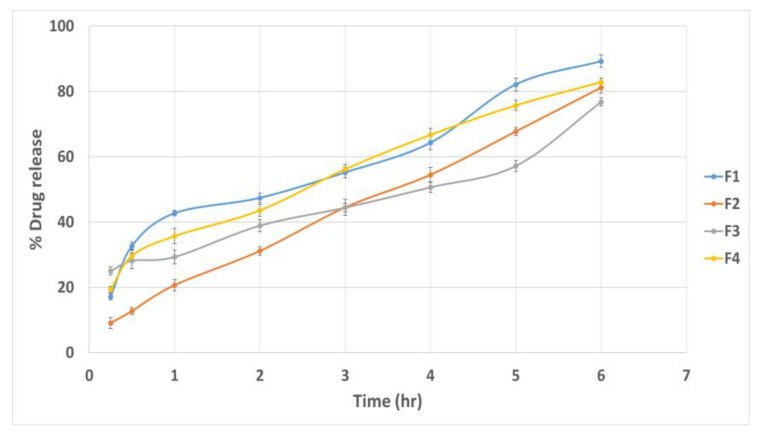
In vitro drug release profiles from different quatsomal gel formulae.

**Figure 11 pharmaceutics-15-01947-f011:**
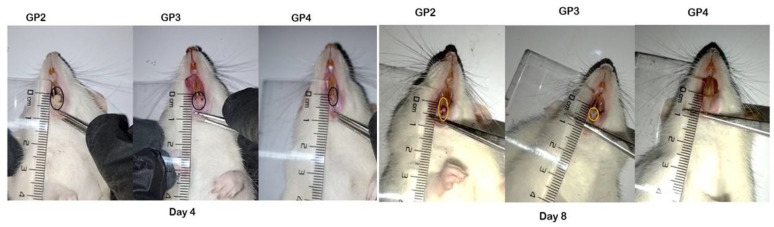
Photographs showing ulcer diameters in different groups after 4 and 8 days, GP2—positive control group, GP3—free drug gel group, GP4—quatsomal gel group; the circles in the photographs point to the site of ulceration in the buccal mucosa.

**Figure 12 pharmaceutics-15-01947-f012:**
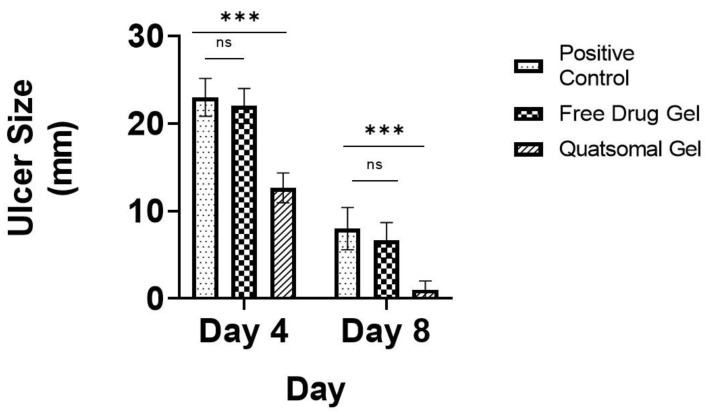
Diameter of ulcers in different groups after 4 and 8 days; ns—not-significant; *** *p* < 0.001.

**Figure 13 pharmaceutics-15-01947-f013:**
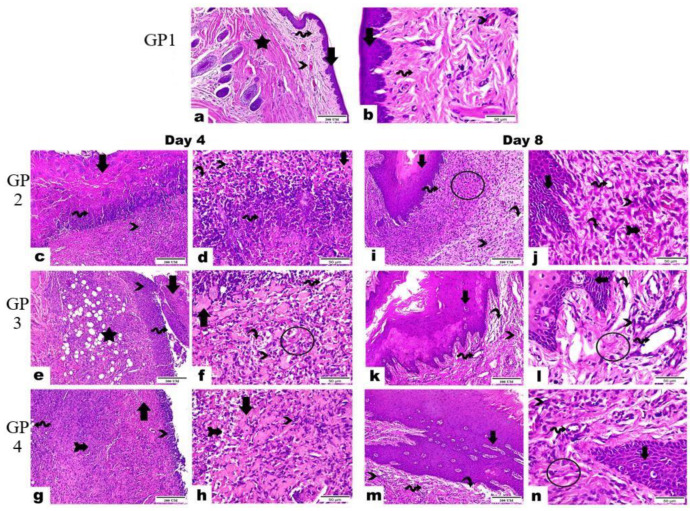
Photomicrographs represented the histopathological differences in buccal tissues among examined groups in 4 and 8 days (Hematoxylin and Eosin stain, magnification power 400×, and scale bar 200 µm and 50 µm), GP1—negative control group, GP2—positive control group, GP3—free drug gel group, GP4—quatsomal gel group.

**Figure 14 pharmaceutics-15-01947-f014:**
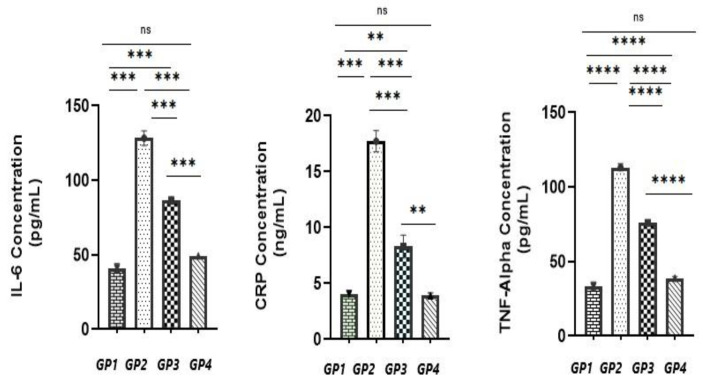
Assessment of serum interleukin-6 (IL-6), C-reactive protein (CRP), and tumor necrosis factor-alpha of different animal groups, GP1—negative control group, GP2—positive control group, GP3—free drug gel group, GP4—quatsomal gel group—ns, not significant; **** *p* < 0.0001; *** *p* < 0.001; ** *p* < 0.01.

**Table 1 pharmaceutics-15-01947-t001:** The 2^3^ Factorial design for optimization of Pred MS-loaded quatsomes.

Independent Variables	Levels	
	Low	High
X_1_: QAS type	DDAB	CTAB
X_2_: QAS: CHO molar ratio	1:1	1:2
X_3_: Sonication time (min)	10	20
Responses (dependent variables)	Constraints	
Y_1_: PS (nm)	Minimize	
Y_2_: PDI	Minimize	
Y_3_: ZP (mV)	Maximize (absolute value)	
Y_4_: EE (%)	Maximize	
Y_5_: Q6%	Maximize	

Abbreviations: EE%—Entrapment efficiency percent, CTAB—cetyltrimethylammonium bromide, DDAB—Dimethyldioctadecylammonium bromide, Pred MS—Prednisolone sodium metazoate, PS—Particle size, PDI—Polydispersity index, CHO—Cholesterol, Q6%—Percent of the drug released after 6 h, ZP—Zeta potential.

**Table 2 pharmaceutics-15-01947-t002:** Experimental runs, independent variables, and the response of the 2^3^ factorial design.

Formulation Code	QAS Type	QAS: CHO Molar Ratio	Sonication Time (min)	PS (nm) (Y_1_)	PDI (Y_2_)	ZP (mV) (Y_3_)	EE % (Y_4_)	Q6 (%) (Y_5_)
F1	DDAB	1:1	10	113.28 ± 0.79	0.324 ± 0.001	56.9 ± 0.42	98.60 ± 1.22	84.21 ± 2.70
F2	CTAB	1:2	10	91.35 ± 0.29	0.278 ± 0.002	48.6 ± 0.63	84.89 ± 1.71	63.54 ± 2.45
F3	CTAB	1:1	10	95.93 ± 0.67	0.328 ± 0.004	45.15 ± 0.19	90.81 ± 0.74	58.39 ± 1.75
F4	DDAB	1:2	10	100.50 ± 0.74	0.234 ± 0.003	53.4 ± 0.14	95.90 ± 1.35	93.89 ± 1.19
F5	CTAB	1:1	20	86.10 ± 0.26	0.269 ± 0.007	58.1 ± 0.35	88.23 ± 1.52	70.41 ± 2.94
F6	DDAB	1:1	20	89.30 ± 0.45	0.291 ± 0.001	61.2 ± 0.43	97.74 ± 0.83	85.84 ± 2.24
F7	DDAB	1:2	20	69.47 ± 0.41	0.207 ± 0.004	68.1 ± 0.54	94.56 ± 1.16	94.42 ± 2.15
F8	CTAB	1:2	20	75.21 ± 0.34	0.213 ± 0.009	61.3 ± 0.28	79.62 ± 1.44	74.11 ± 2.27

Note: Data is represented as mean ± SD (*n* = 3). Abbreviations: PS—particle size; PDI—polydispersity index; ZP—zeta potential, and EE%—entrapment efficiency percentage.

**Table 3 pharmaceutics-15-01947-t003:** Output data of the 2^3^ factorial design analysis.

Source	PS (nm)	PDI	ZP (mV)	EE (%)	Q6 (%)
*p*-value	<0.0001	<0.0001	<0.0001	<0.0001	<0.0001
X_1_ = A = QAS type	<0.0001	0.0011	0.0004	<0.0001	<0.0001
X_2_ = B = QAS: CHO	<0.0001	<0.0001	<0.0001	<0.0001	<0.0001
X_3_ = C = Sonication time	<0.0001	<0.0001	<0.0001	<0.0001	<0.0001
Adequate precision	36.4477	55.8875	86.7139	69.0594	103.0599
R^2^	0.9924	0.9976	0.9991	0.9981	0.9992
Adjusted R^2^	0.9857	0.9955	0.9983	0.9964	0.9986
Predicted R^2^	0.9695	0.9904	0.9964	0.9924	0.9970
Significant factors	X_1,_ X_2,_ X_3_	X_1,_ X_2,_ X_3_	X_1,_ X_2,_ X_3_	X_1,_ X_2,_ X_3_	X_1,_ X_2,_ X_3_

**Table 4 pharmaceutics-15-01947-t004:** The composition and physical characteristics of different gel formulae.

Gel Formulae	Polymers Type	Polymers Percent (%)	pH	Drug Content (%)	Spreadability (g·cm/s)	MF (dyne/cm^2^)	Viscosity cp	Q6 (%)
G1	Noveon	1%	6.9 ± 0.1	96.8 ± 0.41	6.67 ± 0.78	9417.6 ± 82.09	4286 ± 41.51	90.66 ± 0.49
G2	Noveon	3%	6.7 ± 0.14	97.3 ± 0.92	5.5 ± 0.75	10398.6 ± 90.54	7938 ± 45.76	85.53 ± 0.58
G3	Chitosan	3%	5.8 ± 0.22	94.9 ± 0.45	7.5 ± 0.65	7259.4 ± 118.25	9843 ± 59.75	88.31 ± 0.46
G4	Chitosan	1%	5.8 ± 0.13	93.2 ± 0.61	12.91 ± 0.59	5787.9 ± 105.76	277.3 ± 38.21	92.24 ± 0.36

## Data Availability

The data presented in this study are available on request from the corresponding author.

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
