# Peer review of "Formulation and Evaluation of Prednisolone Sodium Metazoate-Loaded Mucoadhesive Quatsomal Gel for Local Treatment of Recurrent Aphthous Ulcers: Optimization, In Vitro, Ex Vivo, and In Vivo Studies"

_pharmaceutics, 2023, doi:10.3390/pharmaceutics15071947_

Round 1

Reviewer 1 Report

Dear Authors,

The manuscript entitled “Formulation and Evaluation of Prednisolone Metasulfo Benzoate Loaded Mucoadhesive Quatsomal Gel for Local Treatment of Recurrent Aphthous Ulcers: Optimization, In Vitro, Ex Vivo, and In Vivo Studies” describes buccal mucoadhesive gel containing prednisolone metasulfobenzoate (pred-MSB) loaded quatsomes for efficient localized therapy of recurrent aphthous ulcers (RAS). Pred-MSB loaded quatsomes were prepared using adapted method of Dong, D. et al (ref. 25) using quaternary ammonium salt (QAS) such as bromide of cetyltrimethylammonium (CTAB) or dimethyldidodecylammonium (DDAB)  (instead of cetylpyridinium chloride – CPC as in ref. 25) and cholesterol (CHO) in molar ratio QAS:CHO 1:1 and 1:2, as well as 0.2 % w/v of Pred MSB and 10 ml MiliQ water sonificated for some time.  The author used a 2^3 factorial design to address the impact of independent variables QAS type (X1), QAS to CHO molar ratio (X2), and sonication time (X3) and dependent variables were particle size (PS; Y1), polydispersity index (PDI; Y2), zeta potential (ZP; Y3), entrapment efficiency percent (EE%; Y4) and percent of drug released after 19 hours (Q6%: Y5) in order to discover optimal quatsomal formula. The selected formula was then incorporated into different gel bases using Noveon or chitosan polymers in different concentrations (1 or 3%) to prepare a mucoadhesive gel. The physicochemical properties of prepared mucoadhesive gels were determined (pH, drug content, rheological properties, in vitro drug release studies and ex-vivo mucoadhesive force). Finally, in vivo studies were performed on 24 Wistar albino rats. The rats were divided in 4 groups (negative control GP1, positive control GP2, GP3 group treated  with gel containing 0.2% w/w pred-MSB and GP4- group treated with mucoadhesive quatsomal gel).  GP2, GP3 and GP4 groups were subjected to ulceration of the buccal mucosa where a round filter paper, 6 mm in diameter, was soaked in 15 μL of 50% acetic acid and was used to cause aseptic tissue necrosis - as in Hitomi, S. et al. (ref. 40). The wound healing was defined by measuring the reduction in the ulcer area. The reduction of ulcers and inflammatory markers show advantages of quatsomal mucoadhesive gel platform and promising potential for the topical delivery of Pred MSB in managing recurrent aphthous ulcers.

The manuscript is well written and data analysed and is of interest for investigation of RAS treatment. However some corrections should be made.

1st and the major comment on this manuscript is the possible toxicity of studied quatsomes. It seems that this issue is not still seriously studied due to novelty of quatsomes platform for drug delivery but knowing the toxicity of its constituents (hydrophobic quarternary ammonium cations) I believe something more about this issue should be mentioned in this manuscript. It is also not clear why CTAB and DDAB are chosen in the first place for the study instead of CPC for example which is already used in pharmaceutical formulations. Similarly, why toxic bromide instead of chlorides were used?

2nd comment: The reading of the manuscript is somewhat difficult due to chemical ambivalence. According to CAS number (3694-41-5), a generic and sufficient name for the active pharmaceutical compound studied in this work is prednisolone metasulfobenzoate (not metasulfo benzoate). Also it is not always clear that compound used in this work is actually its sodium analogue. This compound is named prednisolone sodium metasulfo benzoate which is also incorrect. The name found online for this compound is prednisolone sodium metazoate (CAS number 630-67-1). All the names of active compounds should be in accordance with some coherent nomination system in the  whole text (including title) and eventually full IUPAC name structures and/or CAS numbers should be provided at the end of the text in order to avoid any misunderstanding. 

3rd comment: The structure of the active compound (pred-MS) as well as cholesterol (CHO) and DDAB and eventually CTAB should be provided in a separate scheme also in order to facilitate text reading and particularly analysis of FTIR spectrum.

Thank you for your attention.

Reviewer 2 Report

I reviewed the research article by Kassem et al regarding the development of a mucoadhesive hydrogel for mucosal application, to treat aphtous ulcers.

Overall, I think that their work is valid. Authors completed a huge amount of work and the design of experiments makes sense, they also included the right control groups. I have just a few comments that can improve the paper and make it more useful for other researchers in the field.

I have attached the pdf of the paper with my comments.

Author Response

Response to Reviewer 1

Dear reviewer, thanks a lot for your careful and excellent revision. Regarding your valuable comments, we have revised the manuscript as the following. Also, all the requested revisions have been highlighted within the manuscript.

  • Regarding the solvents used to prepare the 0.2 % w/v Pred MSB.

Response: 20 mg of the drug, the weighted amount of cholesterol and surfactant were added to 10 ml Mili- Q water so that the final drug concentration was 0.2% w/v. Mentioned in section 2.2

  • Regarding: specifying types of QAS within the method of preparation.

Response: The used QAS either CTAB or DDAB has been mentioned in the revised manuscript (section 2.2)

  • Regarding: the temperature of the magnetic stirrer.

Response: room temperature (mentioned in section 2.2)

  • Regarding ref no 25

Response: the reference for the quatsomes preparation method, and its position have been revised (section 2.2).

  • Regarding: the dilution used during measuring PS, PDI and ZP.

Response: the dilution is 10 folds and has been added to the revised manuscript (section 2.3.1)

  • Regarding the time used while measuring the EE%.

Response: we usually perform initial trials to detect the time at which the free drug is completely separated from vesicles via dialysis. In this study, we found that after 1 hour, the free drug concentration in the dialysis media became constant. This method was performed only at room temperature to decrease the risk of drug leakage from pellets. The time has been mentioned in the revised manuscript (section 2.3.2).

  • Regarding the stability study

Response: in this study, the whole formula containing the encapsulated and free drug was utilized to detect the change in the free drug % and easily detect any change in the EE% during storage. Clarified in the revised manuscript (section 2.8).

  • Regarding missing drug content in the context.

Response: the EE% has been added to the revised manuscript (section 2.8).

  • Regarding the volume of water used during the preparation of the chitosan gel

Response: the volume of water has been mentioned in the revised manuscript (section 2.9)

  • Regarding the pH adjustment of Noveon.

Response: triethanolamine was utilized to adjust the pH of Noveon formulae to reach the target pH( buccal mucosa pH). Mentioned in section 2.9

  • Regarding the status of quatsomes added to the gel.

Response: the quatsomes were added to the gel as purified pellets, which is mentioned in the revised manuscript (section 2.9).

  • Regarding time interval of blood samples

Response: the samples were taken at the end of the experiment, day 8 (has been mentioned in the revised manuscript, section 2.11).

  • Regarding Figure 2a

Response: As per the reviewer’s comment, the figure has been adjusted in the revised manuscript.

  • Regarding combining Figs 1,2,3,4&6 in 1 figure

Response: We apologize for not doing that. The figure will be complicated and hard to follow.

  • Regarding drug content

The required quatsomes pellets were mixed with different gel bases to achieve the final drug amount (0.2%w/w). However, to ensure the uniformity of pellets within the performed gel, we measured the actual drug amount and compared it to the theoretical amount (0.2%w/w). Check section 3.11.4.

  • Regarding figure 11

Response: As per the reviewer’s comment, the figure has been revised in the manuscript.

  • Regarding figure 13

Response: As per the reviewer’s comment, the figure has been revised in the manuscript.

Reviewer 3 Report

The present article (pharmaceutics-2470189) refers to the authors' work on the formulation and evaluation of 3-[2-[(8S,9S,10R,11S,13S,14S,17R)-11,17-dihydroxy-10,13-dimethyl-3-oxo-7,8,9,11,12,14,15,16-octahydro-6H-cyclopenta[a]phenanthren-17-yl]-2-oxoethoxy]carbonylbenzenesulfonic acid (prednisolone metasulfobenzoate, Pred MSB) loaded mucoadhesive quatsomal gel for the local treatment of recurrent aphthous ulcers. In vitro, ex vivo and in vivo studies were conducted using the aforementioned Pred MSB formulations. Optimization, in vitro, ex vivo, and in vivo studies were conducted.

The authors conclude that the quatsomal mucoadhesive gel they developed are promising platforms for the topical delivery of Pred MSB in managing recurrent aphthous ulcers.  The manuscript is concisely written and the results well documented. Some points that need to be addressed in the revised article are:

1. On what grounds the authors suggest that the F4 formulation has an optimal release profile, ant not, for example F5. The experimental design softwares are useful tools, but do not take into account the discrete stereoelectronic features of each individual drug compound. Indicatively, the Pred MSB's release is facilitated by the fact that the -SO2OH group is dissociated, at pH=6.8, and this parameter is not included in the experimental design predictions. Moreover, the physicochemical characteristics of the different formulants used in formulations, are also not taken into account. Hence, please comment on the differences of Pred MSB's release from the developed formulations, based on the aforementioned factors and not just on the experimental design results.

2. In the FTIR discussion section (lines 443-456), there is indication of the -SO2OH absorbance peaks.

Mutatis mutandis, the manuscript is of interest to the cognizant reader. Last, albeit the fact that the quality of the English language is satisfactory, the revised version should be proof-read by a native English speaking person.

Albeit the fact that the quality of the English language is satisfactory, the revised version should be proof read by a native English speaking person.

Author Response

Response to Reviewer 2

Dear reviewer, thanks a lot for your careful and excellent revision. Regarding your valuable comments, we have revised the manuscript as the following. Also, all the requested revisions have been highlighted within the manuscript.

  • Regarding your valuable comment concerning in vitro release, this section was rewritten. Please check the revised manuscript on page 11, section 3.6. (Effect of Formulation Variables on the In Vitro Drug Release).

  • Regarding your valuable comment concerning FTIR, this section was rewritten. Please check the revised manuscript on page 13, section 3.9. (Compatibility Assessment of the Optimized Formula Using Fourier Transform Infrared (FTIR)).

Round 2

Reviewer 1 Report

Dear Editor,

The authors corrected and have responded my comments favourably so I believe manuscript can be accepted for publishing after some minor corrections: e.g. graphical quality of some figures should be improved as they are blurred.

Thank you for your attention.

Ps I find it very rude and unprofessional that authors did not respond with letter addressing referee’s comments. I advise them to always include the letter in the future and recommend to editor not to accept revisions without this letter. After all good scientific discussion, exchange of ideas and argumentation is what provides good work published in journal with high IF as Pharmaceutics.